# Does the Expression of Vascular Endothelial Growth Factor (VEGF) and Bcl-2 Have a Prognostic Significance in Advanced Non-Small Cell Lung Cancer?

**DOI:** 10.3390/healthcare11030292

**Published:** 2023-01-18

**Authors:** Marina Markovic, Slobodanka Mitrovic, Aleksandar Dagovic, Dalibor Jovanovic, Tomislav Nikolic, Anita Ivosevic, Milos Z. Milosavljevic, Radisa Vojinovic, Marina Petrovic

**Affiliations:** 1Department of Internal Medicine, Faculty of Medical Sciences, University of Kragujevac, 34000 Kragujevac, Serbia; 2Department of Medical Oncology, University Clinical Center Kragujevac, 34000 Kragujevac, Serbia; 3Department of Pathology, Faculty of Medical Sciences, University of Kragujevac, 34000 Kragujevac, Serbia; 4Department of Pathology, University Clinical Center Kragujevac, 34000 Kragujevac, Serbia; 5Department of Oncology, Faculty of Medical Sciences, University of Kragujevac, 34000 Kragujevac, Serbia; 6Clinic for Nephrology and Dyalisis, University Clinical Center Kragujevac, 34000 Kragujevac, Serbia; 7Clinic for Rheumatology and Allergology, University Clinical Center Kragujevac, 34000 Kragujevac, Serbia; 8Department of Radiology, University Clinical Center Kragujevac, 34000 Kragujevac, Serbia; 9Department of Radiology, Faculty of Medical Sciences, University of Kragujevac, 34000 Kragujevac, Serbia; 10Pulmonology Clinic, University Clinical Center Kragujevac, 34000 Kragujevac, Serbia

**Keywords:** adenocarcinoma (ACs), squamous cell carcinoma (SCCs), non-small cell lung carcinoma (NSCLC), bcl-2, VEGF

## Abstract

Lung cancer is the most common cause of mortality from malignant tumors worldwide. The five-year survival rate for people with advanced stages varies considerably, from 35.4% to 6.9%. The angiogenic potential of bcl2 is not well known, nor is the way in which tumor cells with excessive bcl2 expression affect VEGF production. Hypothetically, given that tumor growth, progression and metastasis are dependent on angiogenesis, the antiapoptotic effect is expected to form a link between these two molecules. The aim of this study was to evaluate the relationship between bcl-2 and VEGF expression, clinicopathological features and survival in 216 patients with advanced NSCLC. Archival tumor tissues were examined by immunohistochemistry for the expression of bcl-2 and VEGF. Immunoreactivity for bcl-2 was observed in 41.4% of NSCLCs, 51% of squamous and 34.8% of adenocarcinomas-expressed Bcl-2. There was an inverse correlation of mononuclear stromal reaction and bcl-2 expression in adenocarcinoma (*p* < 0.0005). A total of 71.8% NSCLCs were VEGF positive, 56% of squamous and 82.2% of adenocarcinomas. High level of VEGF expression was significantly associated with histology type (*p* = 0.043), low histology grade (*p* = 0.014), clinical stage IV (*p* = 0.018), smoking history (*p* = 0.008) and EGFR mutations (*p* = 0.026). There was an inverse correlation in the expression of Bcl-2 and VEGF in NSCLC patients (*p* = 0.039, r = −0.163). Two-year survival of patients with unresectable NSCLC was 39.3%, and 50% of patients were alive at 17 months. Our results demonstrated no difference in survival for patients in advanced NSCLC grouped by bcl-2 and VEGF status. Additionally, we observed an inverse correlation in the expression of Bcl-2 and VEGF in NSCLC and mononuclear reaction and bcl-2 expression in adenocarcinomas.

## 1. Introduction

Lung cancer is the most common cause of morbidity and mortality from malignant tumors worldwide. Five-year survival rate for non-small cell lung cancer is approximately 26.3%. The five-year survival rate for people with more advanced stages of the disease varies considerably, from 35.4% in locally advanced to 6.9% in metastatic lung cancer, which depends not only on the stage, but also on the biological characteristics of the tumor [1]. The fundamental assumption of the modern age of lung cancer research is that diagnosis, prediction of prognosis and prediction of patient therapy can be improved by combining standard clinical parameters (performance status, tumor size, status of lymph nodes and distant metastases, differentiation, etc.), with genetic or biochemical properties of the tumor.

The best-studied markers, which today represent the standard in the diagnosis of LC, are TTF-1 and p63, with high specificity and sensitivity in differentiating squamous from primary and secondary adenocarcinomas [2,3].

The research results show, besides diagnostic, prognostic significance as well; p63 is a favorable prognostic factor in patients with squamous cell carcinoma [4], overexpression of TTF-1 prolongs the overall survival of adenocarcinoma patients both in the early and advanced stages of the disease [5], while Multivariate analysis by Svarton M and associates defines these immunohistochemical markers as the only significant parameters for progression-free survival and overall survival rates in patients with NSCLC treated with erlotinib [6].

The identification of new molecular markers, which could predict the response of the tumor to the applied treatment, can enable the optimization of the treatment for each patient.

Many genes and proteins involved in the regulation of the cell cycle, the process of angiogenesis and apoptosis have been defined as markers that have a significant role in the therapeutic response and clinical outcome in NSCLC patients [7,8,9]. Angiogenesis plays a pivotal role in tumor growth and metastatic spread, as such was validated as an independent prognostic factor [10].

One of the most important and strongest stimulators of tumor angiogenesis is VEGF. In the process of competition between proangiogenic and antiangiogenic factors, an angiogenic trigger occurs, which results in an increase in VEGF production. This complex signaling pathway allows VEGF to exert a number of effects in neoangiogenesis. Although VEGF expression is characteristic mainly of endothelial cells, high expression has been found in various tumor types including lung tumor cells.

This increased expression of VEGF in tumor tissue is the result of many factors, the most important of which is hypoxia, which stabilizes and increases the expression of the transcription factor HIF-1α (Hypoxia-inducible factor-1α), which stimulates the transcription of VEGF, which is secreted, diffuses through the tissue, reaches the endothelial cells and binds to specific receptors on their surface. Some studies show that genetic polymorphism of VEGF correlates with sensitivity, prognosis and therapeutic response of NSCLC patients [11,12].

Apoptosis is an important mechanism for maintaining the homeostasis of the organism. Bcl-2 (B-cell lymphoma 2) is the first discovered oncogene. It was initially discovered in B-cell lymphomas where in 85% of cases it is expressed as a result of translocation of the 14th and 18th chromosomes. This protein is a member of a gene family that also includes proapoptotic genes, and their mutual relationship determines whether proapoptotic or antiapoptotic signals will prevail in the cell [13,14].

Overexpression of the bcl-2 gene was also observed in certain solid tumors such as neoplasms of the colon (90%), gastrointestinal tract (60%), prostate carcinoma (30%), non-small cell lung carcinoma (20%) and melanoma [15,16]. In these tumors, the expression of Bcl-2 is a good prognostic indicator, which seems like a paradox considering that its main function is the inhibition of apoptosis [17,18]. On the other hand, Bcl-2 expression is associated with tumor progression, including the occurrence of liver metastases in colorectal carcinoma and lymphovascular invasion in triple-negative breast carcinoma [19,20,21,22].

The angiogenic potential of bcl2 is not well known, nor is the way in which tumor cells with excessive bcl2 expression affect VEGF production. Hypothetically, given that tumor growth, progression and metastasis are dependent on angiogenesis, the antiapoptotic effect is expected to form a link between these two molecules.

For this reason, the goal of our study is examined the expression of VEGF and Bcl-2 in unresectable non-small cell lung cancer, the association of the expression with the pathohistological characteristics of the tumor as well as the correlation with the clinical parameters of the examined patients.

## 2. Materials and Methods

This clinical-experimental, non-interventional study was conducted in the Center for Oncology and Radiology, the Service for Pathological-Anatomical Diagnostics and the Pulmonary Clinic of the University Clinical Center Kragujevac, in the period from 2017 to 2020.

### 2.1. Study Population

The study population included 216 patients with locally advanced and metastatic non-small cell lung cancer who were diagnosed and treated at the Kragujevac University Clinical Center in the period from 2016 to 2018. Patients with non-small cell squamous and adenocarcinoma of the lung who were classified into unresectable IIIa, IIIb and IV clinical stage based on clinical stage were included in the research. Staging was completed according to the AJCC (American Joint Committee on Cancer) and UICC (The Union for International Cancer Control) criteria, and the histological classification and grading of tumors according to WHO (World Health Organization) recommendations [23,24]. Patient survival was monitored for 24 months in day hospitals, at regular outpatient check-ups and by telephone contact.

To participate in the study patients were required to have the following criteria: at least 18 years of age, normal hepatic, renal and hematological function and an Eastern Cooperative Oncology Group (ECOG) performance status of 0 or 1. Before entering the study, patients underwent a medical history evaluation and physical examination. Patients did not receive either chemotherapy or radiotherapy. Tumor measurements of lesions were assessed by imaging techniques such as computed tomography (CT) of the chest and abdomen, magnetic resonance imaging (MRI) of the brain. Tumor response was assessed according to the Response Evaluation Criteria in Solid Tumors guidelines Radiological assessments were performed by CT scans of the chest and abdomen after every two cycles of chemotherapy. Written informed consent was obtained from each patient prior to the start of the study.

The exclusion criteria were as follows: the existence of malignancy in other locations, histologically proven microcellular lung cancer, resectable NSCLC (I, II and resectable IIIa clinical stage), the patient’s general condition (ECOG performance status 3–4) and the decision of the Lung Oncology Council on treatment of patients using symptomatic therapy or only palliative radiation therapy due to comorbidities that exclude the use of chemotherapy. The confounding variables are the gender and age of the respondents, professional occupation, physical activity and data related to personal and family history.

### 2.2. Method

Tissue samples obtained by bronchoscopy and video-assisted thoracoscopic surgery were used in the research. Standard, histomorphological H&E method and immunohistochemistry were used as methods. Personal and demographic data of patients (gender, age, occupation, habits, comorbidities, family history, etc.), results of radiological diagnostics, data on applied treatment protocols as well as results of EGFR analysis obtained by PCR method in local reference laboratories were taken from the History of Diseases.

#### 2.2.1. H&E Method

On routine, H&E-stained preparations of tumor samples, classic parameters of NSLC were determined by microscopic analysis: histological type, histological and nuclear grade, mitotic index, degree of necrosis and stromal-mononuclear reactions.

#### 2.2.2. Immunohistochemical Method

The 3-micron-thick tissue sections, pre-fixed for 24 h in 10% neutral buffered formalin and embedded in paraffin blocks, were mounted on highly adherent slides, deparaffinized and rehydrated. After epitope release and endogenous peroxidase blocking, the tissue preparations were incubated with primary mouse monoclonal antibodies VEGF (clone VG1; dilution 1:100) and Bcl-2 (clone 100/D5, dilution 1:50), after washing with the secondary antibody, at room temperature according to the manufacturer’s recommendations (ThermoFisher Scientific, Waltham, MA, USA). The reaction was visualized with 3-diaminobenzidine tetrachloride (DAB), counterstained with Mayer’s hematoxylin, and final mounting and coverslipping with Canada balsam. The quality of the immunohistochemical reaction was monitored by simultaneous testing of internal and external, negative and positive tissue controls. The prepared immunohistochemical preparations were analyzed under a light microscope (Axioskop 40, Carl Zeiss, Oberkochen, Germany), and representative fields were photographed with a digital camera (AxioCam ICc1, Carl Zeiss, Oberkochen, Germany). All immunohistochemical stainings were performed with external control of the quality and specificity of staining, using positive and negative controls according to the propositions of NordiQC (Nordic Immunohistochemical Quality Control).

### 2.3. Immunohistochemical Detection of VEGF and Bcl-2 in NSCLC

Immunohistochemical analysis was performed independently by two researchers, without familiarization with the clinical data of the studied population. In the case of preparations with different obtained values, the final result was reached by consensus. Cytoplasmic expression of VEGFa was determined by a semiquantitative method, as a product of the percentage of positive tumor cells and the intensity of the reaction, according to the recommendations of Klein et al. [25]. Percentual expression, i.e., the number of VEGF positive per 100 counted tumor cells, was classified into four categories: 0—absence of immunoreactivity, 1 <30%, 2 31–60% and 3 >61% immunoreactivity, while the staining intensity was graded as follows: 0—no staining, 1—weak; 2—mild and 3—strong staining. By multiplying these values, VEGF expression in all NSLC samples was scored from 1 to 9. Specifically defined, the cutoff point to distinguish low from high VEGF expression was 25% of positive carcinoma cells [26].

Bcl-2 expression was determined as a percentage of positive tumor cells, not taking into account the intensity of the reaction. Sections were considered positive for Bcl-2 when ≥10% of tumor cells were stained in the cytoplasm [27].

Tumor proliferation index was determined as the number of nuclei expressing Ki-67 per 100 counted tumor cells, i.e., it is expressed as a percentage. Based on the obtained values, all NSLC were classified into two groups: those with low proliferative capacity, in which Ki-67 was less than 30%, and the group of NSLC with high Ki-67 index (greater than 30%).

### 2.4. Statistical Analysis

Complete statistical analysis of the data will be performed in the statistical computer program, PASW Statistics, version 26. All continuous variables will be presented as the median. Given the normality of the distribution, differences in the mean values of continuous variables were tested using the Mann-Whitney and Kruscal-Wallis tests for independent samples. Correlation between variables will be examined using Spearman’s correlation. Differences in survival were tested using the Kaplan-Mayaer curve and the Log-rank test. All analyzes will be evaluated at a statistical significance level of *p* ≤ 0.0524.

## 3. Results

### 3.1. Clinical Pathological Characteristics

There were 151 males and 65 females, with an average age of 64 years (range 36–80). The main patient characteristics are shown in Table 1.

### 3.2. Association between the IHC Expression of Bcl-2 and Clinicopathological Features

Immunoreactivity for bcl-2 was observed in 41.4% of NSCLCs. Fifty-one percent of squamous cell carcinomas (SCCs) and 34.8% of adenocarcinomas (ACs) expressed Bcl-2. Immunostaining was localised predominantly in cytoplasm. High Bcl-2 expression was significantly associated with histology grade and stromal-mononuclear reaction (Figure 1). No statistically significant difference was identified between Bcl-2 expression and other clinical parameters, including gender, smoking status, TNM stage or lymph node metastasis (results not shown).

### 3.3. Association between the IHC Expression of VEGF and Clinicopathologial Features

A total of 71.8% patients had positive immunohistochemical staining for VEGF. Fifty-six percent of squamous cell carcinomas (SCCs) and 82.2% of adenocarcinomas (ACs) expressed VEGF. High level of VEGF expression was significantly associated with histology type, histology grade, lymph node status, clinical stage and smoking history (Figure 2).

### 3.4. Relationship between the Expression of VEGFa and Bcl-2

There was an inverse correlation (Figure 3) in the expression of Bcl-2 and VEGF (*p* = 0.039) in NSCLC patients.

### 3.5. Association of the Proliferation Index with the Expression of VEGFa and Bcl-2

A high Ki-67 proliferative index was observed in 79 patients NSCLC (71.8%) and was more common in adenocarcinomas than in squamous cell carcinomas. We found positive correlation between VEGF and the Ki-67 proliferation index in NSCLC patients. No correlation has been seen between Bcl-2 and the Ki-67 proliferation index in NSCLC patients, but there was a negative correlation in adenocarcinoma subgroups (Figure 4).

### 3.6. Association of EGFR Status with Expression of VEGFa and Bcl-2

Wild-type EGFR was present in 86.3%, while 13.7% of adenocarcinoma had mutations on exons 19 and 21 (Table 2).

EGFR status significantly depends on the stage of the disease and age; younger patients more often had a mutated status, not related to histological grade and Bcl-2 expression. Mutation carriers are more often male and non-smokers. There is a significant association between EGFR status and VEGFa expression in the analyzed adenocarcinoma samples (Table 3).

### 3.7. Survival

Two-year survival of patients with unresectable NSCLC was 39.3%, and 50% of patients were alive at 17 months. Survival is statistically significantly affected by age and stage of the disease (results not shown), and without significance, survival is associated with the histological type of tumor, Ki-67 tumor proliferative capacity, VEGF and Bcl-2 expression (Figure 5).

## 4. Discussion

Several experimental studies have shown a connection between bcl2 over expression and increased VEGF production, i.e., the influence of antiapoptotic bcl2 status on growth, progression, metastasis and intensification of angiogenesis [28,29,30]. However, in NSCLC, this relationship is not sufficiently clear, because it has been shown that the expression of bcl2, as a favorable prognostic factor, can be inversely correlated with VEGF [31,32] but also positively correlated with negative prognostic implication [33,34].

In our study, we examined the expression of VEGF and Bcl-2 in the tissue of 216 patients with locally advanced and metastatic NSCLC, association of IHC expression and clinicopathologial features, as well as correlation of IHC expression.

Stefanoup et al., reported that the expression of VEGF was 77.3% in NSCLC [35]. In this study, expression rate of VEGF protein was 71.8%.

The statistical analysis showed that positive expression of VEGF was higher in well differentiated cancer, in adenocarcinoma, lymph node metastasis and clinical stage of IV. The results in the present study is consistent with previous reports [36,37,38].

Usuda et al. showed in their research that VEGF expression is significantly higher in adenocarcinoma compared to squamous cell carcinomas, and that VEGF can be a positive regulator of pericytes and angiogenesis, thus affecting the proliferation of endothelial cells and maturation of blood vessels of tumor, which could result in a favorable outcome in well-differentiated lung cancer [39].

Lymphnodal metastases play an important role in tumor staging, which depends on the therapeutic modality, and as such represent an important prognostic predictor. In addition to being associated with tumor size, histological type and lymph node metastases are also associated with VEGF expression [40]. Liu et al. examined the expression of VEGF-C in NSCLC patients and showed that lymphovascular density is higher in VEGF-C positive tumors, which means that VEGF-C can affect both tumor growth and the occurrence of lymphnodal metastases [41]. The association between VEGF expression and lymphnodal metastases has also been shown in other solid tumors such as esophageal, prostate and stomach cancers [42,43,44].

We found statistically significant association between VEGF expression in current smokers and never smokers. In vitro findings are consistent with a pro-angiogenic effect of nicotine; increased angiogenesis in response to nicotine and the role of nicotine in tumorigenesis has not been demonstrated. There are some evidence indicate that prostacyclin is involved in angiogenesis and it is known that nicotine releases prostacyclin from human vascular endothelial cells. Also, nicotine induces basic fibroblast growth factor and platelet-derived growth factor release in endothelial cells and increases proliferation [45,46]. Heeschen C et al. have shown that nicotine increases endothelial cell number, reduces apoptosis and increases capillary network formation in vitro. The effects of nicotine on vascular structure are mediated by non-neuronal nicotine-sensitive acetylcholine receptors nAChR which have been shown to be present on endothelial cell which is associated with functional changes in tissue blood flow and accelerating tumor growth [47]. Conklin et al. show nicotine and cotinine cause an increase in VEGF expression in endothelial cells, indicating that cotinine may have an even more pronounced effect than nicotine because of the long half-life of cotinine [48]. Some studies suggested that nicotine causes a significant increase in serum VEGF levels [47,49,50,51]. Zhao H et al. showed that a low dose of nicotine promoted VEGF secretion by increasing HIF-1α-mediated VEGF transcription under hypoxic conditions [52].

The results of our study show a significant association between VEGF expression and EGFR (*p* = 0.026). In preclinical studies, mutations of EGFR correlated with VEGF in lung cancer, as it has been shown that EGFR-mutant NSCLC cells hasten the expression of VEGF more than wild-type NSCLC cells [53]. The reason for this is that EGFR-mutant NSCLC cells constitutively up-regulate HIF-1a in a hypoxia-independent manner and HIF-1a induces VEGF expression [54,55].

Some studies have shown that at the time of detection of resistance to the application of tyrosine kinase inhibitor (TKi), resistant cells had an elevated level of VEGF, thus suggesting the mutual connection of EGFR and VEGF signaling pathways. Based on this assumption, numerous clinical studies have shown benefit from dual EGFR-VEGF signaling inhibition, including longer time to disease progression and resistance to TKi [56,57,58].

In addition, we indicate that VEGF expression was significantly positively correlated with Ki-67 expression. These results indirectly show that high VEGF expression assumes a poorer prognosis of the disease.

Yoshihito Shibata et al., confirmed Bcl-2 expression in 29.1% of NSCLC and 44.4% of squamous cell carcinomas [27]. The results of this and other studies are similar to ours in terms of a higher number of squamous cell Bcl-2 positive tumors compared to the number of Bcl-2 positive NSCLC. Expression of Bcl-2 was related to histology grade and mononuclear stromal reactions (*p* = 0.022, *p* < 0.0005). Our results show that of bcl-2 over expression may represent tumor de-differentiation and more aggressive behaviour of NSCL tumors.

Petrisor et al. showed that Bcl-2 expression is lower in poorly differentiated colon cancers compared to low-grade tumors [59]. The assumption is that during tumor progression, Bcl-2 expression decreases due to its inhibitory effect on the progression of the cell cycle, thus leading to a reduction in the inhibition of apoptosis [60]. No survival difference was detected based on Bcl-2 status. Krug et al. also reported no difference in survival for patients in advanced NSCLC grouped by bcl-2 status [61].

The impact of bcl-2 protein expression on prognosis in NSCLC patients shows controversy. The meta-analysis by Zhang et al. shows the association of bcl-2 positivity and LONGER survival, while on the other hand some studies come to the conclusion that bcl-2 expression has no prognostic significance or is even associated with a poor prognostic outcome [62,63].

What could explain this are different antibodies used for immunohistochemical analysis, cut-off values, sample size and tumor heterogeneity. Bcl-2 expression may be more informative if put into context with the expression of other family members with pro-apoptotic functions. Published results suggest that the high expression of a pro-apoptotic Bax protein was associated with poor prognosis in esophageal, squamous cell carcinoma, Bak in bladder cancer and Mcl-1 protein defined as unfavorable prognostic marker for lung cancer [64,65,66,67].

According to Gurova et al., loss of apoptosis through p53 inactivation leads to genetic instability and tumor progression, whereas loss of apoptosis through bcl-2 overexpression creates genetically stable tumors that escape the selective pressure to inactivate p53 and are therefore less prone to progression, leading to a better prognostic outcome [68]. Haldar et al. show negative regulation of Bcl-2 expression by some p53 mutants that can result in a decrease in expression of Bcl-2 in advanced tumors with mutated p53 [69].

An inverse correlation was found between expression of Bcl-2 and VEGF. Michael I. Koukourakis et al. and Fontanini et al. show inverse relationship between vascular grade and bcl-2 expression but in operable NSCLC [70,71]. It has been shown that p53 oncoprotein can regulate neovascularisation through induction of thrombospondin-1, an inhibitor of angiogenesis. On the other hand, bcl-2 may inhibit p53 function that can be associated with increased tumoral angiogenesis. Therefore, bcl-2 may regulate angiogenesis either by stimulating the release of inhibitors of angiogenesis or by suppression of angiogenic factors via a pathway distinct from that of p53.

## 5. Conclusions

In conclusion, the present study demonstrated no difference in survival for patients in advanced NSCLC grouped by bcl-2 and VEGF status. Additionally, we observed a inversely correlation in the expression of Bcl-2 and VEGF in NSCLC patients and inverse correlation of mononuclear stromal reaction and bcl-2 expression in adenocarcinomas.

## Figures and Tables

**Figure 1 healthcare-11-00292-f001:**
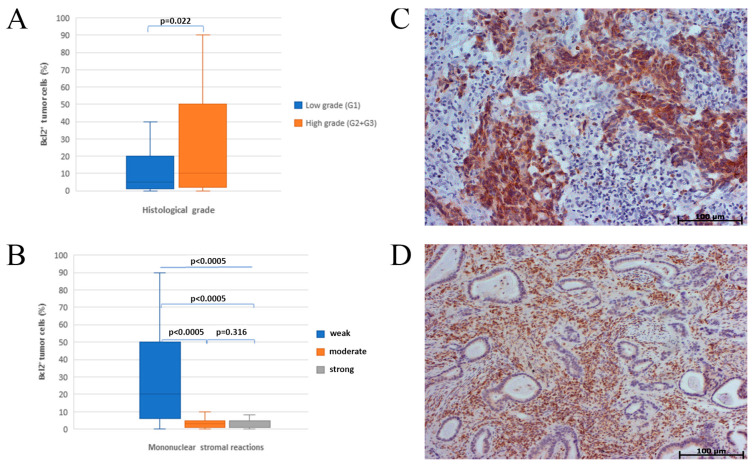
Association of Bcl-2 expression with histological grade and mononuclear stromal reaction; (**A**) influence of histological grade on bcl-2 expression; histomorphological presentation of positive correlation of bcl-2 expression and histological grade (**C**); inverse correlation of mononuclear stromal reaction and bcl-2 expression in adenocarcinoma (**B**); histomorphological representation of the intensity of mononuclear stromal reaction and bcl-2 expression in adenocarcinoma (**D**).

**Figure 2 healthcare-11-00292-f002:**
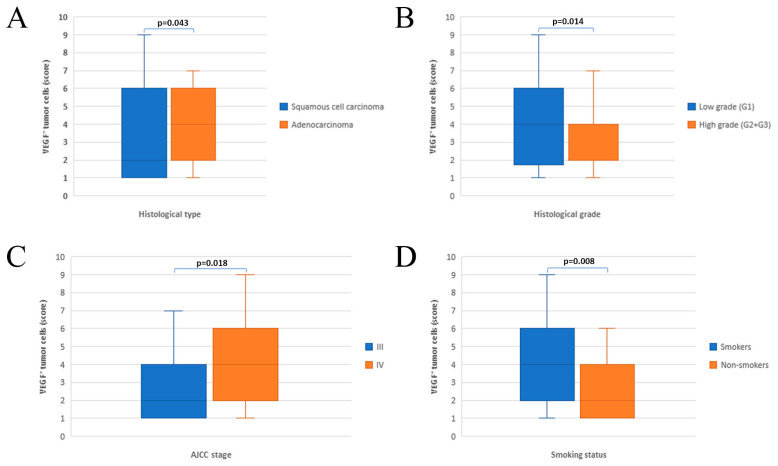
Assotiation between the IHC expression of VEGF and clinicopathiological features (**A**) histological type; (**B**) histological grade; (**C**) stage disease; (**D**) smoking status.

**Figure 3 healthcare-11-00292-f003:**
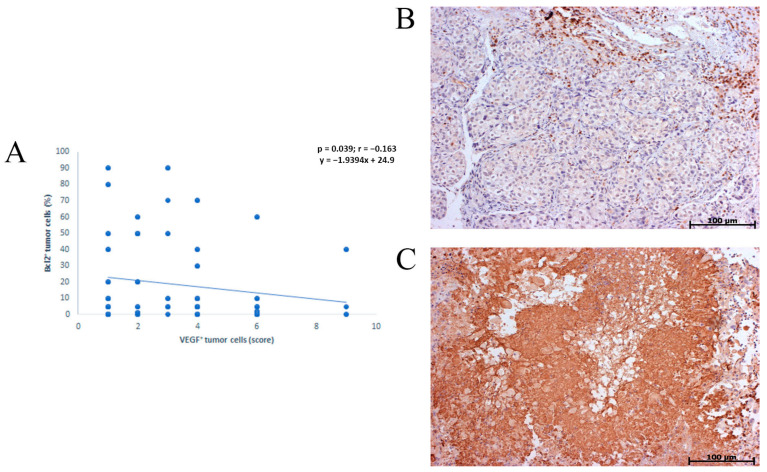
Graphic and histomorphological presentation of inverse Correlation between Bcl-2 and VEGF expression in NSCLC. (**A**) graphic of inverse corelation; (**B**,**C**) histomorphological presentation.

**Figure 4 healthcare-11-00292-f004:**
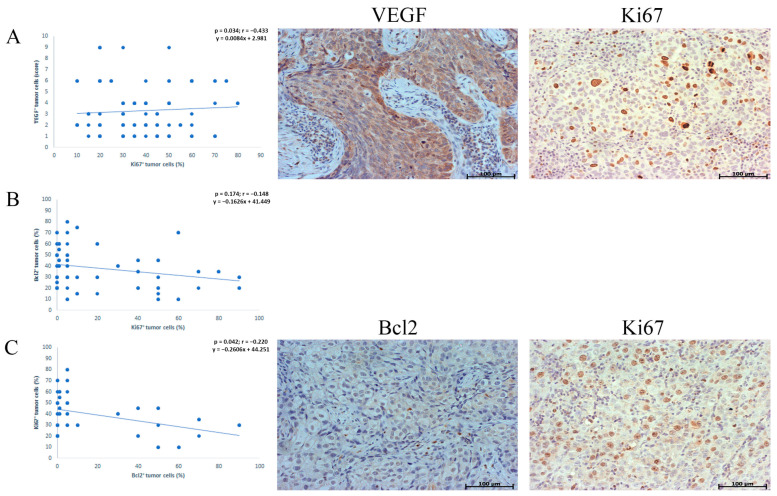
Association of the proliferation index with the expression of VEGFa and Bcl-2. (**A**) graphic and histomorphological presentation of positive correlation between VEGF and the Ki-67 proliferation index in NSCLC patients; (**B**) relationship between Ki-67 proliferative index and bcl-2 expression in NSCLC patients (**C**) graphic and histomorphological presentation of inverse correlation between Bcl-2 and the Ki-67 proliferation index in adenocarcinomas.

**Figure 5 healthcare-11-00292-f005:**
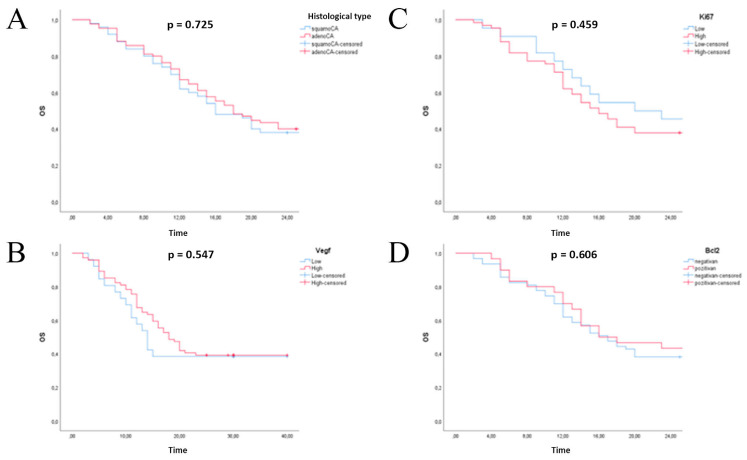
Kaplan-Meier overall survival analysis of patients with non-small lung cell carcinoma. (**A**) histology type; (**B**) expression status of VEGF; (**C**) expression status of Ki67; (**D**) expression status of Bcl-2.

**Table 1 healthcare-11-00292-t001:** Baseline Characteristics of the Study Population of Advance NSCLC Patients.

Patient and Tumor Characteristics	Number of Cases	%
**Sex**		
Male	151	69.9
Female	65	30.1
**Histology**		
Squamous	81	37.5
Adenocarcinoma	135	62.5
**Histology grade**		
Low grade	34	16.6
High grade	172	83.4
**Smoking**		
Current smoker	176	85.4
Never smoked	30	14.6
**Tumor size**		
T2	22	10.2
T3	43	20.0
T4	150	69.8
**Node status**		
N0	5	2.3
N1	24	11.1
N2	142	65.7
N3	45	20.8
**Stage**		
III	71	32.8
IV	145	67.2

**Table 2 healthcare-11-00292-t002:** Percentage of adenocarcinoma mutations.

		EGFR	
		Frequency	Percent
**Valid**	wild	88	86.3
	mutations	14	13.7
	total	102	100.0

**Table 3 healthcare-11-00292-t003:** Correlation between EGFR and VEGF and Bcl-2 in 102 cases of advanced lung adenocarcinoma.

Variables	Bcl-2				VEGF		
		n	q	*p*-Value	n	q	*p*-Value
**EGFR**	Wild	1	(0.40)	0.802	3	(2.4)	0.026
	mutations	10	(0.10)		4	(4.6)	

## Data Availability

The data presented in this study are available on request from the corresponding author.

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
