# Peer review of "Does the Expression of Vascular Endothelial Growth Factor (VEGF) and Bcl-2 Have a Prognostic Significance in Advanced Non-Small Cell Lung Cancer?"

_healthcare, 2023, doi:10.3390/healthcare11030292_

Round 1

Reviewer 1 Report

Marina Markovic et al conducted an interesting study evaluating the relationship between bcl-2 or VEGF expression, and clinicopathological features and survival respectively in patients with advanced NSCLC.

This paper is in large lines well directed, the subject being of real interest to the scientific community. However, I consider that there are some minor issues and concerns that the authors should address.

- The introduction explains the scientific background and justification of the investigation. However, it would be preferable, in addition to presenting the specific objectives, to include any pre-determined hypotheses - if they exist. 

-In the methods section, please provide the eligibility criteria for participants.

-Within the methods, please clearly define all outcomes, exposures, predictors and potential confounders.

-Tables 2 and 3 should be revised according to the requirements of the journal

-In addition to Kaplan-Meier, consider using a Cox regression to investigate the effect of variables on survival

-In order to provide readers with an overview of prognosis in advanced non-small cell lung cancer, I strongly suggest that the authors should include in their discussion the prognostic significance of other markers (such as p63 or ) that play a role in predicting outcome (e.g.: https://doi.org/10.3892/ol.2020.11663 ; doi: 10.1111/1759-7714.12181. ; doi: 10.7150/jca.26830) as well as tumor differentiation (eg: https://pubmed.ncbi.nlm.nih.gov/31263838/ ; https://pubmed.ncbi.nlm.nih.gov/15188024/.)

Author Response

Reviewer 1

1) ”The introduction explains the scientific background and justification of the investigation. However, it would be preferable, in addition to presenting the specific objectives, to include any pre-determined hypotheses - if they exist.“

The angiogenic potential of bcl2 is not well known, nor is the way in which tumor cells with excessive bcl2 expression affect VEGF production. Hypothetically, given that tumor growth, progression, and metastasis are dependent on angiogenesis, the antiapoptotic effect is expected to form a link between these two molecules. (This issue is included in the last part of the Introduction section.)

2)”In the methods section, please provide the eligibility criteria for participants.”

To participate in the study patients were required to have the following criteria: at least 18 years of age, normal hepatic, renal and hematological function, an Eastern Cooperative Oncology Group (ECOG) performance status of 0 or 1. Before entering the study, patients underwent a medical history evaluation and physical examination. Patients did not receive either chemotherapy or radiotherapy. Tumor measurements of lesions were assessed by imaging techniques such as computed tomography (CT) of the chest and abdomen, magnetic resonance imaging (MRI) of the brain.   Written informed consent was obtained from each patient prior to the start of the study. (This issue is included in the  Materials and Methods, 2.1. Study population section).

3)”Within the methods, please clearly define all outcomes, exposures, predictors and potential confounders.”

The outcome of the disease was monitored through 24-month survival monitoring as part of regular clinical examinations and through telephone contact.

The exclusion criteria were: the existence of malignancy in other locations, histologically proven microcellular lung cancer, resectable NSCLC (I, II and resectable IIIa clinical stage), the patient's general condition (ECOG performance status 3-4) and the decision of the Lung Oncology Council to treatment of patients using symptomatic therapy or only palliative radiation therapy due to comorbidities that exclude the use of chemotherapy. The confounding variables are the gender and age of the respondents, professional occupation, physical activity and data related to personal and family history.

(This issue is included in the  Materials and Methods, 2.1. Study population section).

4) ”Tables 2 and 3 should be revised according to the requirements of the journal.”

Thanks for noticing the omission, the tables were revised according to the journal's propositions.

5)”In addition to Kaplan-Meier, consider using a Cox regression to investigate the effect of variables on survival.”

The Omnibus Tests of Model Coefficients table shows the analysis of the validity of the regression models that included the histological type and the expressions of VEGF, Bcl2 and Ki67. Preliminary statistics have determined that such regression models do not have statistical validity, as can be seen from the probability values in the same table. Because of this result, in the paper we only presented the analysis of survival using the Log-rank test and the Kaplan-Meier curve, which did not confirm differences in survival depending on the examined parameters. The table titled Variables in the Equation lists the remaining parameters of the Cox regression model. Please, see supplement 1.

6)”In order to provide readers with an overview of prognosis in advanced non-small cell lung cancer, I strongly suggest that the authors should include in their discussion the prognostic significance of other markers (such as p63 or ) that play a role in predicting outcome (https://doi.org/10.3892/ol.2020.11663  doi: 10.1111/1759-7714.12181. ; doi: 10.7150/jca.26830) as well as tumor differentiation (eg: https://pubmed.ncbi.nlm.nih.gov/31263838/ ; https://pubmed.ncbi.nlm.nih.gov/15188024/.)”

We are grateful for the good idea on how to improve the quality of our work. We have studied the recommended literature and we believe that an additional description and analysis of the diagnostic and prognostic significance of p63 and TTF-1 markers fits best in the introduction of our research, because it is difficult to find a discussion connection with the examined  antiapoptotic bcl2 and angiogenic VEGF markers. However, if the reviewer thinks that a more suitable place for this supplement is the discussion section in the paper, we will try to respect his idea. (This issue is discussed in line 55-65 of the Introduction section.)

Reviewer 2 Report

The manuscript titled 'Does the expression of vascular endothelial growth factor (VEGF) and Bcl-2 have a prognostic significance in advanced non-small cell lung cancer?' attempts to examine the expression profiles for VEGF and Bcl-2 in NSCLC and associate them with the tumor pathological characteristics to make clinical associations. While the authors attempt to draw conclusions based on sampling tissues from patients suffering from different types and stages of NSCLC, the evaluated criteria and approach are not necessarily novel. Multiple previously published papers have adopted a similar approach and validated the increased expression of VEGF and Bcl-2 in NSCLC patients. In fact, the authors have cited many of these papers. In the eyes of this reviewer, the manuscript lacks originality or novelty in experimental design. Additionally, the current findings have been established in the field for some time and do not necessarily provide advancement to the field. Other minor concerns include a lack of details within the methods section, an Illegible figure key, a lack of scale bars on figure images, no clear logical connection established to explore VEGF and Bcl-2 within the manuscript, limited discussion for the caveats associated with the experimental design and the adopted approach. 

Author Response

Reviewer 2

1)”Multiple previously published papers have adopted a similar approach and validated the increased expression of VEGF and Bcl-2 in NSCLC patients. In fact, the authors have cited many of these papers. In the eyes of this reviewer, the manuscript lacks originality or novelty in experimental design. Additionally, the current findings have been established in the field for some time and do not necessarily provide advancement to the field.”

Numerous earlier studies have shown that overexpression of bcl2 with its antiapoptotic properties increases the degree of proliferation, enhances the migratory, invasive and metastatic potential of tumors. [1,2]. Given that the growth and progression of tumors are dependent on the degree of neovascularization, it is logical that anti-apoptosis can be associated with the creation or increase of an angiogenic phenotype. [3-5]. Experimental in vitro and in vivo analyzes have undoubtedly demonstrated this, reporting a positive correlation between bcl2 overexpression and VEGF molecule production in breast, prostate, neuroblastoma, and melanoma tumors. [3-6]. However, the results of clinical research show that this connection is insufficiently clear and much more complicated, and that probably the genotypic heterogeneity of the tumor determines the activation of other intracellular signaling pathways that would regulate the expression of VEGF independently of the bcl2 status. [7,8,9]. This is supported by the fact that the expression of VEGF is not always correlated with the microvascular density of the tumor. [10-12]. Also, the results of clinical research, especially in the field of lung cancer, show that overexpression of bcl2 can be inversely correlated with VEGF as a favorable prognostic factor [13-15], but also positively correlated with a negative prognostic implication. [16, 17].  Bearing in mind the huge amount of data from experimental research that bcl2 and VEGF can potentially be a promising therapeutic target in the induction of apoptosis and prevention of neoangiogenesis [18,19,20] and that the results of clinical research are insufficiently clear and subject to debate, we think that our idea on the analysis of these molecules, their relationship and impact on disease prognosis is absolutely justified.

1.Zörnig, M.;  Hueber, A.;  Baum, W.; Evan, G., Apoptosis regulators and their role in tumorigenesis. Biochim Biophys Acta 2001, 1551 (2), F1-37.

2.Antonsson, B.; Martinou, J. C., The Bcl-2 protein family. Exp Cell Res 2000, 256 (1), 50-7.

3.Beierle, E. A.;  Strande, L. F.; Chen, M. K., VEGF upregulates Bcl-2 expression and is associated with decreased apoptosis in neuroblastoma cells. J Pediatr Surg 2002, 37 (3), 467-71.

4.Biroccio, A.;  Candiloro, A.;  Mottolese, M.;  Sapora, O.;  Albini, A.;  Zupi, G.; Del Bufalo, D., Bcl-2 overexpression and hypoxia synergistically act to modulate vascular endothelial growth factor expression and in vivo angiogenesis in a breast carcinoma line. FASEB J 2000, 14, 652–660.

5.Iervolino, A.;  Trisciuoglio, D.;  Ribatti, D.;  Candiloro, A.;  Biroccio, A.;  Zupi, G.; Del Bufalo, D., Bcl-2 overexpression in human melanoma cells increases angiogenesis through VEGF mRNA stabilization and HIF-1-mediated transcriptional activity. Faseb j 2002, 16 (11), 1453-5.

6.Fernandez, A.;  Udagawa, T.;  Schwesinger, C.;  Beecken, W.;  Achilles-Gerte, E.;  McDonnell, T.; D'Amato, R., Angiogenic potential of prostate carcinoma cells overexpressing bcl-2. J Natl Cancer Inst 2001, 93 (3), 208-13. 

7.Nabors, L. B.; Gillespie, G. Y.; Harkins, L.; King, P. H., HuR, a RNA stability factor, is expressed in malignant brain tumors and binds to adenine- and uridine-rich elements within the 3' untranslated regions of cytokine and angiogenic factor mRNAs. Cancer Res 2001, 61 (5), 2154-61.

8.Gagnon, M. L.;  Bielenberg, D. R.;  Gechtman, Z.;  Miao, H. Q.;  Takashima, S.;  Soker, S.; Klagsbrun, M., Identification of a natural soluble neuropilin-1 that binds vascular endothelial growth factor: In vivo expression and antitumor activity. Proc Natl Acad Sci U S A 2000, 97 (6), 2573-8.

9.Jiménez, B.;  Volpert, O. V.;  Crawford, S. E.;  Febbraio, M.;  Silverstein, R. L.; Bouck, N., Signals leading to apoptosis-dependent inhibition of neovascularization by thrombospondin-1. Nat Med 2000, 6 (1), 41-8.

10.Hirschi, K. K.;  Rohovsky, S. A.;  Beck, L. H.;  Smith, S. R.; D'Amore, P. A., Endothelial cells modulate the proliferation of mural cell precursors via platelet-derived growth factor-BB and heterotypic cell contact. Circ Res 1999, 84 (3), 298-305.

11.Nör, J. E.;  Mitra, R. S.;  Sutorik, M. M.;  Mooney, D. J.;  Castle, V. P.; Polverini, P. J., Thrombospondin-1 induces endothelial cell apoptosis and inhibits angiogenesis by activating the caspase death pathway. J Vasc Res 2000, 37 (3), 209-18.

12.Padró, T.;  Ruiz, S.;  Bieker, R.;  Bürger, H.;  Steins, M.;  Kienast, J.;  Büchner, T.;  Berdel, W. E.; Mesters, R. M., Increased angiogenesis in the bone marrow of patients with acute myeloid leukemia. Blood 2000, 95 (8), 2637-44.

13.Fontanini, G.;  Boldrini, L.;  Vignati, S.;  Chinè, S.;  Basolo, F.;  Silvestri, V.;  Lucchi, M.;  Mussi, A.;  Angeletti, C. A.; Bevilacqua, G., Bcl2 and p53 regulate vascular endothelial growth factor (VEGF)-mediated angiogenesis in non-small cell lung carcinoma. Eur J Cancer 1998, 34 (5), 718-23.  

14.Bairey, O.;  Zimra, Y.;  Shaklai, M.; Rabizadeh, E., Bcl-2 expression correlates positively with serum basic fibroblast growth factor (bFGF) and negatively with cellular vascular endothelial growth factor (VEGF) in patients with chronic lymphocytic leukaemia. Br J Haematol 2001, 113 (2), 400-6.  

15.Jinyoung Yoo, J. H. J., Hyun Joo Choi, Seok Jin Kang and Chang Suk Kang, Expression of bcl-2, p53 and VEGF in Non-Small Cell Lung  Carcinomas: Their Relation with the Microvascular Density and  Prognosis. The Korean Journal of Pathology 2005, 39, 74-80.

16.Ahmed, M. B.;  Nabih, E. S.;  Louka, M. L.;  Abdel Motaleb, F. I.;  El Sayed, M. A.; Elwakiel, H. M., Evaluation of nestin in lung adenocarcinoma: relation to VEGF and Bcl-2. Biomarkers 2014, 19 (1), 29-33.   

17.Tian, J.;  Hu, L.;  Li, X.;  Geng, J.;  Dai, M.; Bai, X., MicroRNA-130b promotes lung cancer progression via PPARγ/VEGF-A/BCL-2-mediated suppression of apoptosis. J Exp Clin Cancer Res 2016, 35 (1), 105.

18.Rosen, L. S., Clinical experience with angiogenesis signaling inhibitors: focus on vascular endothelial growth factor (VEGF) blockers. Cancer Control 2002, 9 (2 Suppl), 36-44.  

19.Zeitlin, B. D.;  Zeitlin, I. J.; Nör, J. E., Expanding circle of inhibition: small-molecule inhibitors of Bcl-2 as anticancer cell and antiangiogenic agents. J Clin Oncol 2008, 26 (25), 4180-8.

20.Vilenchik, M.;  Raffo, A. J.;  Benimetskaya, L.;  Shames, D.; Stein, C. A., Antisense RNA down-regulation of bcl-xL Expression in prostate cancer cells leads to diminished rates of cellular proliferation and resistance to cytotoxic chemotherapeutic agents. Cancer Res 2002, 62 (7), 2175-83.

2) “Other minor concerns include a lack of details within the methods section,….”

Thanks for the suggestion. This issue is included in the  Materials and Methods, 2.1. Study population section.

3) “…. an  Illegible figure key, ….”

Respecting the recommendations, a description of the figures can be found in the text. However, if necessary, we can send more detailed legends of the figures.

4) “….a lack of scale bars on figure images,….”

The scale bars are included in the figures.

5)”….no clear logical connection established to explore VEGF and Bcl-2 within the manuscript, limited discussion for the caveats associated with the experimental design and the adopted approach.” 

The angiogenic potential of bcl2 is not well known, nor is the way in which tumor cells with excessive bcl2 expression affect VEGF production. Hypothetically, given that tumor growth, progression, and metastasis are dependent on angiogenesis, the antiapoptotic effect is expected to form a link between these two molecules. (This issue is included in the last part of the Introduction section).

Several experimental studies have shown a connection between bcl2 over expression and increased VEGF production, i.e. the influence of antiapoptotic bcl2 status on growth, progression, metastasis and intensification of angiogenesis [28, 29, 30]. However, in NSCLC, this relationship is not sufficiently clear, because it has been shown that the expression of bcl2, as a favorable prognostic factor, can be inversely correlated with VEGF [31,32] but also positively correlated with negative prognostic implication [33, 34]. (This issue is included in the first part of the Discussion section).

Round 2

Reviewer 2 Report

The authors have addressed the concerns raised in the first round of reviews. 

Author Response

Thank you for your review.

Best regards,

Slobodanka
